# In Vitro Antimicrobial Susceptibilities of *Francisella tularensis* subsp. *holarctica* Isolates from Tularemia Outbreaks That Occurred from the End of the 20th Century to the 2020s in Spain

**DOI:** 10.3390/antibiotics10080938

**Published:** 2021-08-03

**Authors:** Sonia Martínez-Martínez, Elías-Fernando Rodríguez-Ferri, David Rodríguez-Lázaro, Marta Hernández, José-Ignacio Gómez-Campillo, María del Carmen Martínez-Nistal, María-Isabel Fernández-Natal, María-José García-Iglesias, Olga Mínguez-González, César-Bernardo Gutiérrez-Martín

**Affiliations:** 1Departmento de Sanidad Animal, Facultad de Veterinaria, Universidad de León, Campus de Vegazana s/n, 24007 León, Spain; smarm@unileon.es (S.M.-M.); ef.rferri@unileon.es (E.-F.R.-F.); mjgari@unileon.es (M.-J.G.-I.); 2Unidad de Microbiología, Departamento de Biotecnología y Ciencia de los Alimentos, Facultad de Ciencias, and Research Centre for Emerging Pathogens and Global Health, University of Burgos, 09001 Burgos, Spain; drlazaro@ubu.es; 3Laboratorio de Biología Molecular y Microbiología, Instituto Tecnológico Agrario de Castilla y León (ITACyL), 47071 Valladolid, Spain; hernandez.marta@gmail.com; 4Laboratorio Regional de Sanidad Animal, Villaquilambre, 24193 León, Spain; gomcamjo@jcyl.es (J.-I.G.-C.); MARNISCA@jcyl.es (M.d.C.M.-N.); 5Jefe de Servicio de Análisis Clínicos/Coordinadora del Laboratorio de Diagnóstico Clínico del Complejo Asistencial Universitario, Hospital Universitario de León, 24071 León, Spain; ifernandez@saludcastillayleon.es; 6Jefe de Servicio de Sanidad Animal, Consejería de Agricultura, Ganadería y Desarrollo Rural, Dirección General de Producción Agropecuaria, Junta de Castilla y León, 47014 Valladolid, Spain; MinGonOl@jcyl.es

**Keywords:** *Francisella tularensis* subsp. *holarctica*, tularemia, hare, vole, tick, human, antimicrobial susceptibility, Spain

## Abstract

A collection of 177 *Francisella tularensis* subsp. *holarctica* clinical isolates (29 from humans and 148 from animals, mainly hares and voles) was gathered from diverse tularemia outbreaks in the Castilla y León region (northwestern Spain) that occurred from the end of the 20th century to the 2020s. Along with four *F. tularensis* subsp. *holarctica* reference strains, all of these clinical isolates were tested using a broth microdilution method to determine their susceptibility to 22 antimicrobial agents, including β-lactams, aminoglycosides and one member each of the tetracycline, glycylcycline, quinolone and sulphonamide classes. Many multi-resistance profiles were found among the tested isolates, but especially among those of human origin (all but two isolates showed resistance to at least 13 of 18 antimicrobial agents). Even so, all human isolates were susceptible to gentamicin and tobramycin, while more than 96% of animal isolates were susceptible to these two aminoglycosides. Ciprofloxacin showed activity against more than 92% of animal and human isolates. However, almost 21% of human isolates were resistant to tetracycline, and more than 65% were resistant to tigecycline. Finally, a quite similar activity to other *F. tularensis* subsp. *holarctica* isolates collected 20 years earlier in Spain was observed.

## 1. Introduction

*Francisella tularensis* is an aerobic Gram-negative intracellular bacterium that it is found in nature in association with a wide variety of wild animals. It is the etiological agent of tularemia, a rare acute zoonosis [1]. Lagomorphs are the most common source of human infection and ticks are the main arthropod vectors, but large airborne and water-transmitted plagues have also been reported [2]. *F. tularensis* comprises three subspecies: *F. tularensis* subsp. *tularensis* (or type A) is found in North America and is the most virulent; *F. tularensis* subsp. *holarctica* (or type B) is found in Europe, Japan and North America and is less virulent, and *F. tularensis* subsp. *mediasiatica* causes a type B-like illness in rabbits [3].

*F. tularensis* is considered as a feasible weapon for use in bioterrorism, as an extremely low dose (about 10^1^–10^2^ CFU) is enough to cause disease [4]. For this reason, it is essential to identify effective antibiotics for the treatment of tularemia. It is known that proper therapy often resolves the infection [5]. Traditionally, aminoglycosides and tetracyclines have been used against tularemia, and since the end of the 20th century and the first half of the 2010s, fluoroquinolones have been suggested as acceptable alternatives for inhibiting in vitro *F. tularensis* growth [6,7,8].

The aim of this study was to ascertain the in vitro antimicrobial susceptibilities of 177 *F. tularesis* subsp. *holarctica* isolates recovered from the end of the 20th century to the 2020s from Castilla y León (northwest of Spain), a geographic location where several tularemia outbreaks were recorded in those two decades, and to compare these antibiotic activities with that of another set of *F. tularensis* subsp. *holarctica* field isolates recovered from tularemia outbreaks at the end of the 20th century [8].

## 2. Results

### 2.1. Antimicrobial Sensitivity to 148 Animal F. tularensis Isolates

The minimum inhibitory concentration (MIC) range, MIC_50_, MIC_90_, and rate of susceptibility of 148 animal *F. tularensis* subsp. *holarctica* field isolates to 22 antimicrobial agents are shown in Table 1. The MIC values for the two control reference strains (described in Section 4) were in the expected range (data not shown).

Within the β-lactams, ampicillin was tested singly and in combination with sulbactam, but very low activity, equal to or lesser than 2.0%, was achieved in both cases. No *F. tularensis* subsp. *holarctica* isolates were susceptible to piperacillin supplemented with tazobactam, with a tailing MIC distribution of between >16/4 and >128/4 mg/L. Ticarcillin was mixed with clavulanic acid, and the activity rate was the highest for these penicillins combined with β-lactamase inhibitors; even so, only 15.1% of the isolates were susceptible.

Carbapenems were represented by ertapenem and meropenem, and monobactams by aztreonam. These three β-lactams were barely effective, with at least 76.3% (113 out of 148) of isolates being resistant to them. Antibiotics covering four generations of cephalosporins were also tested among β-lactams. Activities lower than 6% were found for cephalothin and cefazolin, two first-generation cephalosporins. Cefuroxime, a second-generation compound, showed an even lower activity of 2.7% (4 out of 148 isolates). Ceftriaxone and cefpodoxime, representatives of third-generation cephalosporins, were ineffective against most *F. tularensis* subsp. *holarctica* isolates, with both showing activity of 8.1%, while ceftazidime was effective against only 38.5% of isolates (57 out of 148). The only fourth-generation cephalosporin, cefepime, did not show enhanced activity, with efficiencies lower than 4%. Cefotoxin was the only antibiotic that was not interpreted because a breakpoint (see definition in Table 1; Table 2) for this antibiotic could not be found. The MIC range was clearly unimodal (detailed data not shown), with values from <4 to >32 mg/L. The fact that the MIC range, MIC_50_, and MIC_90_, were similar to those of cefuroxime, the other second-generation cephalosporin tested, seems to indicate that most *F. tularensis* subsp. *holarctica* isolates would also be resistant to cefoxitin (Table 1).

Apart from β-lactams, the three tested aminoglycosides (amikacin, gentamicin and tobramycin) were extremely effective; at least 96.6% of the *F. tularensis* subsp. *holarctica* isolates were inhibited by some of these agents, and gentamicin showed an MIC_50_ and MIC_90_ of <2 mg/L. High levels of activity were also recorded for tetracycline, a well-known broad-use antibiotic, with almost 95% of isolates being susceptible. Similarly, ciprofloxacin proved to be a highly active fluoroquinolone against more than 92% of *F.*
*tularensis* subsp. *holarctica* isolates. Tigecycline behaved somewhat differently; only 76.4% of isolates were susceptible to this glycylcycline. Considerably lower activity was found for the combination of sulfamethoxazole and trimethoprim, with, at most, 23% of isolates (34 out of 148) being regarded as susceptible to this combination (Table 1).

### 2.2. Antimicrobial Sensitivity to 29 Human F. tularensis Isolates

The MIC range, MIC_50_, MIC_90_, and rate of susceptibility of 29 human *F. tularensis* subsp. *holarctica* clinical isolates to 22 antimicrobial agents are shown in Table 2. The MIC results for the two other human *F. tularensis* subsp. *holarctica* reference strains (described in Section 4) were not different from those of the human clinical isolates (data not shown).

Globally, the resistance percentages shown by human and animal isolates against the 22 agents tested were quite similar and sometimes the same, such as for piperacillin and tazobactam, aztreonam, cephalothin and cefepime (Table 1; Table 2). In contrast, there were considerable differences among seven agents depending on whether the isolates had an animal or human source. Specifically, the activity of ertapenem against animal isolates was 23.7% (35 out of 148) and against human isolates it was 6.9% (2 out 29). The rate of susceptibility for animal *F.*
*tularensis* subsp. *holarctica* isolates was 38.5% (57 out of 148), and the rate decreased to 10.4% (3 out of 29) for human strains. With regard to aminoglycosides, the frequency of susceptibility to amikacin was almost 90.0% (26 out of 29 human isolates). All human clinical isolates were susceptible to both tobramycin and gentamycin, while low resistance was found among the animal isolates. The percentage of *F. tularensis* subsp. *holarctica* isolates that were susceptible to tetracycline was 94.6% (140 out of 148) for animal sources to 79.3% (23 out of 29) for human sources. When tigecycline was tested with human isolates, only about one-third of them were found to be susceptible (Table 2). Quite the opposite, it was a highly active fluoroquinolone against more than 92% of human isolates.

### 2.3. Statistical Analysis

The five antibiotics showing the highest susceptibility rates (three aminoglycosides, tetracycline and ciprofloxacin) were selected in order to verify whether there were significant differences according to the time frame, the geographical location (Burgos, Palencia, Valladolid and Zamora, the Castilla y León provinces where the highest levels of isolates were recovered) or the isolation source in which the greatest recoveries were achieved (humans, hares and voles). Significant differences were seen between the results obtained for amikacin and tetracycline and the time frame (*p* < 0.01 for amikacin and *p* = 0.009 for tetracycline). No significant differences were seen for any of the other associations studied (Table 3).

### 2.4. Resistotypes

Resistotype patterns were expressed according to the antimicrobial agent for which the breakpoint was encountered (for all except cefoxitin). Animal or human *F. tularensis* subsp. *holarctica* isolates were resistant to a group of 6 to 18 compounds (Table 4). The most prevalent animal resistotype was the one in which isolates were resistant to 14 antimicrobial agents (39 isolates, 26.3%), followed by the one in which the bacterium was resistant to 15 agents (35 isolates, 23.6%). However, the most prevalent human resistotype was the one in which isolates were resistant to 16 antimicrobial agents (13 isolates, 44.8%), and 93.2% (27 isolates) were resistant to 13 or more antimicrobials (Table 4).

## 3. Discussion

Standardized in vitro antimicrobial susceptibility testing methods are reliable for indicating in vivo antimicrobial effectiveness [15]. A collection of 177 *F. tularensis* subsp. *holarctica* isolates recovered from humans and several animals was tested for in vitro susceptibility to the antimicrobial agents commonly used to treat diseases caused by this Gram-negative organism.

Antimicrobial susceptibility testing of *F. tularensis* subsp. *holarctica* isolates is of crucial interest for detecting the emergence of resistances to first-line drugs, in which case new therapeutic alternatives should be tried. The antimicrobial classes usually recommended for tularemia treatment are aminoglycosides, tetracyclines and fluoroquinolones [13]. As effective and safe vaccines are not currently available, tularemia prophylaxis relies on the administration of effective antibiotics [16]. In this study, five antibiotics belonging to the three classes cited above, along with another 17 agents, were tested.

Human and animal isolates showed extremely low susceptibility to ampicillin, with MIC_50_ and MIC_90_ values quite in agreement with those documented almost 20 years ago in Spain [8]. Similarly, no isolate susceptible to ampicillin was found in studies carried out in other countries [9,17,18]. Ampicillin and sulbactam, ticarcillin and tazobactam and ticarcillin and clavulanic acid did not improve the activity of ampicillin tested singly (Table 1; Table 2). In any event, the MIC_50_ and MIC_90_ values reported for piperacillin and tazobactam in our study were lower than those detected in our country at the end of the 20th century [8]. Nevertheless, the resistance of *F. tularensis* subsp. *holarctica* isolates to piperacillin and tazobactam remains very high at the beginning of the 21st century in Spain. This combination has been studied by other authors [17,18,19], resulting in MIC_50_ and MIC_90_ values even higher than those in the current study.

With regard to cephalosporins, none of the seven tested compounds (Table 1) were effective against *F. tularensis* subsp *holarctica* isolates, showing 38.5% activity at best. This inactivity is in general agreement with what was reported in Spain two decades ago [8], except for ceftazidime, for which the MIC range, MIC_50_ and MIC_90_ decreased considerably over that time. However, some studies [7,17,18] revealed MIC ranges for ceftazidime and cefepime much higher than those detected in our research. A Turkish study [18] showed similar MIC_50_ and MIC_90_ values to those reported in our investigation for these five cephalosporins. However, the susceptibility of Japanese isolates showed large variations when tested against cephalosporins [20].

In relation to the other β-lactam antibiotics, aztreonam and the two penems examined in this study again showed low activity rates (Table 1; Table 2) and, once more, our MIC values are comparable to those reported by García del Blanco et al. [8] in Spain. In earlier reports [7,17], even lower activity than ours was shown for meropenem; they obtained MIC ranges of 1.5 to >32 mg/L and >32 mg/L, and our MIC range was <1 to >8 mg/L. The same result was found for aztreonam, with a MIC range of <4 to >32 mg/L versus the >256 mg/L found previously [17].

Conversely, the three aminoglycosides exhibited excellent activity against human and animal *F. tularensis* subsp. *holarctica* isolates. Again, comparing our results and those previously achieved in Spain [8], it can be inferred that there was no increase in resistance in *F. tularensis* subsp. *holarctica* isolates over the 2010s and 2020s in our country. Most related studies have also shown high efficacies for aminoclycosides [9,18,21], especially for amikacin and gentamicin [13,14,17,20,22,23,24,25] and their results are quite similar to those of our study.

Tetracyclines are another type of antibiotics routinely used to treat tularemia. In this context, all studies have documented that tetracycline is highly effective against the etiological agent of tularemia, both previous studies [14,17,18,22,23,25] and ours. Nevertheless, the MIC values found at the end of the 20th century in Spain (MIC range of 4 to 64 mg/L, MIC_50_ of 16 mg/L, and MIC_90_ of 64 mg/L) [8] were quite higher than those seen in the present study.

Glycylcyclines are semisynthetic analogues of tetracyclines that were designed to overcome two of the resistance mechanisms typical of tetracyclines (the efflux pumps and/or the ribosome defenses). Tigecycline, the first member of this antibiotic class, reaches high intracellular concentrations in tissues, macrophages and neutrophils, which makes this agent an interesting alternative for the treatment of disorders caused by intracellular pathogens such as *F. tularensis* subsp. *holarctica* [16,26]. Tigecycline has been shown to be more active than doxycycline against Turkish and French *F. tularensis* [9,13,23]; however, surprising resistance of up to 65.5% was detected in our research among human isolates. These striking in vitro data are not indicative of the potential usefulness of tigecycline in tularemia patients, at least for infections caused by the isolates recovered in Spain, in contrast to the findings of other studies [14,16,18].

Fluoroquinolones are another group chosen as the first line of defense in the routine treatment of tularemia. Only ciprofloxacin was included in the GN3F microtiter panel used in this study, and *F. tularensis* isolates demonstrated susceptibility in 92.6% of animal and 92.4% of human isolates. Similar activity triggered by ciprofloxacin [17,18,20,23,27] and other fluoroquinolones (such as gatifloxacin, grepafloxacin, levofloxacin, moxifloxacin and trovafloxacin) has been reported [12,14,16,18,22]. Even so, a higher MIC range was found for these 177 clinical strains than for those assessed at the end of the 20th century in Spain [8], and this might imply that a certain degree of resistance to ciprofloxacin could have emerged in *F. tularensis* subsp. *holarctica* from 1999 [8] to the years covered in this study. Finally, the combination of sulfamethoaazole and trimethoprim has rarely been tested against this pathogen. Based on the resistance found among the 177 tested isolates, its use in the field must be strongly discouraged.

Since no significant differences were obtained in the susceptibility of isolates to amikacin, gentamicin, tobramycin, tetracycline and ciprofloxacin when comparing isolation sources, Castilla y León provinces and especially isolation periods, it can be inferred that the resistance of *F. tularensis* subsp. *holarctica* isolates to these five antibiotics did not increase from 2007 to the 2020s. This means that the same antibiotics that were active several years ago continue to be effective. We are unable to find an explanation for the reduction from 100 to 75% in susceptibility to amikacin in the periods 1997–1998 and 1999–2006, or for the new increment back to 100% from 2007 to 2020 (*p* < 0.01) (Table 3) beyond the smaller number of isolates collected in some of the established frames. The same can be said for the increases and decreases seen for tetracycline from 1997 to 2020 (*p* = 0.009) (Table 3).

The results shown in Table 4 clearly demonstrate that all of the tested *F. tularensis* subsp. *holarctica*, especially from human sources, displayed resistances to many of the antimicrobial agents evaluated, that is to say, they showed multiresistance patterns. However, a comparison with other isolates could not be carried out because no resistotypes have been previously depicted. In any case, this is the first study to date in which such a large amount of data on the multiresistance of *F. tularensis* subsp. *holarctica* is reported, at least in Spain. The fact that our isolates were effectively inhibited by the three antibiotic classes commonly used for the treatment of human tularemia suggests that the outlook for the future is not as daunting as was feared.

## 4. Materials and Methods

### 4.1. Francisella tularensis Strains

A total of 150 animal *F. tularensis* subsp. *holarctica* strains were used in this study. Of these, 148 isolates were recovered from tularemia outbreaks that occurred from 2007 to 2020 in Castilla y León (most in Palencia, Valladolid or Zamora provinces), which is both the largest autonomous community in Spain and the largest region in all of Europe. Most came from hares and voles (95 and 38, respectively) and, to a lesser extent, from ticks (n = 8), field mice (n = 5), wild rabbits (n = 1), and shrews (n = 1). These isolates had previously been characterized biochemically and molecularly [27,28,29]. The remaining *F. tularensis* subsp. *holarctica* were two reference strains (CAPM 5536 and CAPM 5537), whose origin was the former Czech Republic, and both were recovered from hares.

In addition, 31 human *F. tularensis* isolates were also tested. These were recovered from tularemia outbreaks that occurred in Castilla y León (11 from León, 10 from Zamora and 5 each from Palencia and Valladolid) from 1998 to 2019 and were characterized similarly [27,28,29]. In addition, two other reference strains from human sources were examined: CAPM 5600 (Schu strain), isolated in the USA in 1941, and NTCT 10857, the live vaccine strain of *F. tularensis* type B.

Finally, quality control was verified using *Staphylococcus aureus* ATCC 29213, *Pseudomonas aeruginosa* ATCC 27853, and *Escherichia coli* ATCC 25922, according to the Clinical and Laboratory Standards Institute (CLSI) [10,11].

### 4.2. Antimicrobial Sensitivity Testing

The antimicrobial susceptibilities of the human and animal isolates described above were determined by the broth microdilution method using commercially prepared dehydrated-antibiotic 96-well microtiter plates (GN3F^TM^, Sensititre, Thermo Scientific, The Netherlands) according to the recommendations of the CLSI [10,11]. Each isolate was grown aerobically at 37 °C for 2 days on Neisseria Selective Medium Plus (Oxoid, Spain), and a bacterial suspension was made in demineralized water equivalent to a 0.5 McFarland standard. Then, 100 μL of the mixture was added to Mueller-Hinton broth with N-Tris (hydroximethyl)methyl-2-amino-thanesulfonic acid (TES) (Thermo Scientific, The Netherlands) and supplemented with calcium and magnesium ions, 0.1% glucose, and 2% Vitox^TM^ (Oxoid) [8]. After this, the commercial panel was reconstituted by adding 100 μL/well of the inoculum and the plates were incubated at 37 °C for 72 h. The antimicrobial agents and their dilution ranges are shown in Table 1; Table 2. Each panel was read visually and the minimal inhibitory concentration (MIC) was established as the lowest concentration of agent inhibiting the visible growth of the inoculum. Breakpoint was established as the concentration (in mg/L) of an antimicrobial agent which defines whether *F. tularensis* subsp. *holarctica* is susceptible or resistant to that compound. The percentage of resistant isolates could be calculated when the breakpoint for a given antimicrobial agent was available [8,9,12,13].

### 4.3. Statistical Analysis

The associations between the susceptibility of *F. tularensis* subsp. *holarctica* isolates to the five most effective antimicrobial agents and the most common isolation sources (humans, hares and voles), the Castilla y León provinces where the highest number of isolates were obtained (Palencia, Valladolid and Zamora) and the time frames (1997–1998, 1999–2006, 2007–2010, 2014–2015, 2016–2017 and 2019–2020, set up according to the number of isolates found each year) were tested by χ^2^ for independence, and the data were expressed as percentages of susceptibility of the isolates to the antibiotics. SPSS software version 2.4 (SPSS Inc., IBM, Chicago, IL, USA) was used to perform the statistical analysis, and differences were considered significant at *p* < 0.05.

## 5. Conclusions

Most antimicrobial agents were ineffective against both human and animal Spanish *F. tularensis* subsp. *holarctica* isolates in vitro, giving rise to a large number of multiresistance patterns. However, amikacin, gentamicin, tobramycin, tetracycline and ciprofloxacin were highly effective against them, irrespective of their source. In addition, similar susceptibility was detected for these five antibiotics compared to that seen for other *F. tularensis* subsp. *holarctica* isolates collected in Spain from the end of the 20th century to the first half of 2010s.

## Figures and Tables

**Table 1 antibiotics-10-00938-t001:** Minimum inhibitory concentration (MIC) range, MIC_50_, MIC_90_, and percentage of susceptibility of 148 animal *Francisella tularensis* subsp. *holarctica* field isolates to 22 antimicrobial agents.

Antimicrobial Agent (Concentrations Used in mg/L)	MIC Range (mg/L)	MIC_50_ (mg/L)	MIC_90_ (mg/L)	Breakpoint (mg/L) *	Susceptibility of Isolates to Antimicrobial Agents (%)
Ampicillin (4–32)	<4 to >32	>32	>32	≤4 ^a^	2.0
Ampicillin/sulbactam (4–32/2–16)	<4/2 to >32/16	>32/16	>32/16	≤4 ^a^	1.4
Piperacillin/tazobactam (16/4–128/4)	<16/4 to >128/4	<16/4	32/4	≤8 ^b^	0.0
Ticarcillin/clavulanic acid (16/2–64/2)	<16/2 to >64/2	16/2	>64/2	≤16 ^c^	15.5
Ertapenem (2–16)	<2 to >16	>16	>16	≤4 ^d^	23.7
Meropenem (1–8)	<1 to >8	>8	>8	≤4 ^a^	6.1
Aztreonam (4–32)	<4 to >32	>32	>32	≤8 ^b^	7.4
Cephalothin (2–16)	<2 to >16	>16	>16	≤8 ^e^	3.4
Cefazolin (4–32)	<4 to >32	>32	>32	≤8 ^e^	5.4
Cefoxitin (4–32)	<4 to >32	32	>32	---	---
Ceftazidime (1–32)	<1 to 32	1	4	≤1 ^b^	38.5
Ceftriaxone (1–64)	<1 to >64	2	8	≤1 ^b^	8.1
Cefuroxime (4–32)	<4 to >32	>32	>32	≤8 ^b^	2.7
Cefepime (4–32)	<4 to >32	32	>32	≤4 ^b^	3.4
Cefpodoxime (2–16)	<2 to >16	4	>16	≤2 ^c^	8.1
Amikacin (8–64)	<8 to 32	<8	<8	≤16 ^e^	98.0
Gentamicin (2–16)	<2 to 16	<2	<2	≤8 ^f^	98.7
Tobramycin (4–8)	<4 to 8	<4	<4	≤8 ^g^	96.6
Tetracycline (0.5–16)	<0.5 to >16	2	2	≤4 ^f^	94.6
Tigecycline (1–8)	<1 to 8	<1	2	≤2 ^h^	76.4
Ciprofloxacin (0.5–4)	<0.5 to >4	<0.5	<0.5	≤0.5 ^f^	92.6
Sulfamethoxazole/trimethoprim (9.5/0.5–76/4)	<9.5/<0.5 to >76/>4	>76/>4	>76/>4	≤38/2 ^c^	23.0

* Breakpoint: Concentration (in mg/L) of antimicrobial agent that defines whether a given bacterium (*F. tularensis* subsp. *holarctica* in this case) is susceptible or resistant to the compound; ^a^ [9]; ^b^ [10] (for Enterobacteriaceae); ^c^ [11,12] (for Enterobacteriaceae); ^d^ [11,12] (for meropenem); ^e^ [11,12]; ^f^ [13]; ^g^ [13] (for gentamicin); ^h^ [14].

**Table 2 antibiotics-10-00938-t002:** Minimum inhibitory concentration (MIC) range, MIC_50_, MIC_90_, and percentage of susceptibility of 29 human *Francisella tularensis* subsp. *holarctica* clinical isolates to 22 antimicrobial agents.

Antimicrobial Agent (Concentrations Used in mg/L)	MIC Range (mg/L)	MIC_50_ (mg/L)	MIC_90_ (mg/L)	Breakpoint (mg/L) *	Susceptibility of Isolates to Antimicrobial Agents (%)
Ampicillin (4–32)	<4 to >32	>32	>32	≤4 ^a^	2.0
Ampicillin/sulbactam (4–32/2–16)	<4/2 to >32/16	>32/16	>32/16	≤4 ^a^	1.4
Piperacillin/tazobactam (16/4–128/4)	<16/4 to 128/4	<16/4	128/4	≤8 ^b^	0.0
Ticarcillin/clavulanic acid (16/2–64/2)	<16/2 to >64/2	32/2	>64/2	≤16 ^c^	15.1
Ertapenem (2–16)	<2 to >16	>16	>16	≤4 ^d^	23.7
Meropenem (1–8)	<1 to >8	>8	>8	≤4 ^a^	6.1
Aztreonam (4–32)	<4 to >32	>32	>32	≤8 ^b^	7.4
Cephalothin (2–16)	<2 to >16	>16	>16	≤8 ^e^	3.4
Cefazolin (4–32)	<4 to >32	>32	>32	≤8 ^e^	5.4
Cefoxitin (4–32)	<4 to >32	32	>32	---	---
Ceftazidime (1–32)	<1 to 32	2	16	≤1 ^b^	38.5
Ceftriaxone (1–64)	<1 to >64	4	64	≤1 ^b^	8.1
Cefuroxime (4–32)	<4 to >32	>32	>32	≤8 ^b^	2.7
Cefepime (4–32)	<4 to >32	>32	>32	≤4 ^b^	3.4
Cefpodoxime (2–16)	<2 to >16	16	>16	≤2 ^c^	8.1
Amikacin (8–64)	<8 to 64	<8	16	≤16 ^e^	97.9
Gentamicin (2–16)	<2	<2	<2	≤8 ^f^	98.7
Tobramycin (4–8)	<4	<4	<4	≤8 ^g^	96.6
Tetracycline (0.5–16)	0.5 to 8	2	4	≤4 ^f^	94.6
Tigecycline (1–8)	<1 to 8	2	4	≤2 ^h^	76.4
Ciprofloxacin (0.5–4)	<0.5 to2	<0.5	<0.5	≤0.5 ^f^	92.6
Sulfamethoxazole/trimethoprim (9.5/0.5–76/4)	<9.5/<0.5 to >76/>4	>76/>4	>76/>4	≤38/2 ^c^	23.0

* Breakpoint: Concentration (mg/L) of an antimicrobial agent that defines whether a given bacterium (*F. tularensis* subsp. *holarctica* in this case) is susceptible or resistant to the compound; ^a^ [9]; ^b^ [10] (for Enterobacteriaceae); ^c^ [11,12] (for Enterobacteriaceae); ^d^ [11,12] (for meropenem); ^e^ [11,12]; ^f^ [13]; ^g^ [13] (for gentamicin); ^h^ [14].

**Table 3 antibiotics-10-00938-t003:** Results of *p*-values obtained from statistical analysis of five antibiotics showing highest susceptibility percentages and variables “isolation source”, “Castilla y León provinces”, and “time frames” (set up according to the number of isolates existing each year).

Antibiotic Parameters Tested	Rate of Susceptibility (%)	*p*-Value	Antibiotic Parameters Tested	Rate of Susceptibility (%)	*p*-Value
**Amikacin**			**Tetracycline**		
Humans	96.8	0.114	Humans	93.5	0.234
Hares	100		Hares	99.0	
Voles	100		Voles	97.4	
Palencia	100	--- (ns)	Palencia	98.3	0.767
Valladolid	100		Valladolid	96.1	
Zamora	100		Zamora	97.2	
1997–1998	100	**< 0.01 ***	1997–1998	90.0	**0.009 ***
1999–2006	75		1999–2006	100	
2007–2010	100		2007–2010	100	
2014–2015	100		2014–2015	100	
2016–2017	100		2016–2017	86.7	
2019–2020	100		2019–2020	100	
**Gentamicin**			**Ciprofloxacin**		
Humans	100	0.594	Humans	100	0.594
Hares	99.0		Hares	99.0	
Voles	97.4		Voles	97.4	
Palencia	98.3	0.482	Palencia	98.3	0.713
Valladolid	100		Valladolid	98.0	
Zamora	100		Zamora	100	
1997–1998	100	0.679	1997–1998	100	0.392
1999–2006	100		1999–2006	100	
2007–2010	97.1		2007–2010	98.6	
2014–2015	100		2014–2015	100	
2016–2017	100		2016–2017	93.3	
2019–2020	100		2019–2020	100	
**Tobramycin**					
Humans	100	0.446			
Hares	96.9				
Voles	94.7				
Palencia	96.7	0.329			
Valladolid	94.1				
Zamora	100				
1997–1998	100	0.712			
1999–2006	100				
2007–2010	95.7				
2014–2015	97.4				
2016–2017	93.3				
2019–2020	100				

Ns, not significant; * Significant *p*-values are highlighted in red.

**Table 4 antibiotics-10-00938-t004:** Resistotypes found among 148 animal and 29 human *Francisella tularensis* subsp. *holarctica* isolates according to the number of antimicrobial agents to which they showed resistance.

Number of Antimicrobial Agents to Which Resistance of Isolates Was Detected	Number of Animal Isolates (n = 148) (%)	Number of Human Isolates (n = 29) (%)
6	1 (0.7)	1 (3.4)
7	2 (1.4)	-
8	1 (0.7)	1 (3.4)
9	3 (2.0)	-
10	4 (2.7)	-
11	7 (4.7)	-
12	9 (6.1)	-
13	22 (14.9)	1 (3.4)
14	39 (26.3)	3 (10.3)
15	35 (23.6)	4 (13.9)
16	20 (13.5)	13 (44.8)
17	5 (3.4)	4 (13.9)
18	-	2 (6.9)

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
