# Peer review of "In Vitro Antimicrobial Susceptibilities of Francisella tularensis subsp. holarctica Isolates from Tularemia Outbreaks That Occurred from the End of the 20th Century to the 2020s in Spain"

_antibiotics, 2021, doi:10.3390/antibiotics10080938_

Round 1

Reviewer 1 Report

Present manuscript deals with assessment of resistance and susceptibility of Francisella tularensis to different antibacterial agents. The outline presented in the abstract and introduction provides an idea of the results presented in the manuscript. However, the results have not been explained well and are full of erroneous use of words and sentences. The section does not make any point clear. Later sections are a little better represented but needs extensive editing. The authors should use the service of professional editors or the one provided by the antibiotics journal.

Although the authors have chosen to address an important topic, in the present form, the manuscript does not seem to convey the results from their study. It would be an interesting article if presented clearly. I have a few comments given below. I suggest, the authors should prepare their manuscripts properly and get edited before they submit.

In results, the authors need to clearly frame the sentences for readers to be able to follow. For example, an increase in resistance to ampicillin from 100 to 98.6 % is not clear. Also, the use of sulbactam did not improve the resistance is an important point and authors must use alternate ways to address it. Use of another B lactamase inhibitor in these experiments should suffice. Overall, the purpose of this section, the objective and representation of their data in terms of an increase and decrease in the resistance is not clear and I suggest authors to rewrite this sentence after clearly thinking it through. The authors need to clearly mention the purpose and their method of measurement of resistance used in this study. What do the authors mean by a “break point” and why is it necessary?

The authors use of “worse” when less percentage of resistant isolates are obtained does not appear suitable for a scientific literature especially for this Journal which keeps publishing research aimed at reducing the incidents of antibiotic resistance.

Tigecycline appears to be a very efficient drug as only 25 % of the isolates show resistance to it. Any comments?

Line 68 – Penicillin and Ampicillin are two different antibiotics. The authors must specify what they want to convey in the sentence.

Line 71 – “all animal isolates”? Please use complete sentences in the manuscript. It has been mentioned in the beginning of the section that animal isolates are being discussed. The repetition here is confusing. The authors must clearly present results from microorganisms isolated from different sources.

Line 120 – missing “of”

I do not think that the random incidence of drug resistance can be represented as “rate of susceptibility.

Line 175 – Ampicillin was strongly inactive is not an ideal combination

Author Response

ANSWERS TO REVIEWER 1

Present manuscript deals with assessment of resistance and susceptibility of Francisella tularensis to different antibacterial agents. The outline presented in the abstract and introduction provides an idea of the results presented in the manuscript. However, the results have not been explained well and are full of erroneous use of words and sentences. The section does not make any point clear. Later sections are a little better represented but needs extensive editing. The authors should use the service of professional editors or the one provided by the antibiotics journal.

The different sections but especial the results have been rewritten and all our manuscript has been editing in its English by the MDPI English services.

Although the authors have chosen to address an important topic, in the present form, the manuscript does not seem to convey the results from their study. It would be an interesting article if presented clearly. I have a few comments given below. I suggest, the authors should prepare their manuscripts properly and get edited before they submit.

In results, the authors need to clearly frame the sentences for readers to be able to follow. For example, an increase in resistance to ampicillin from 100 to 98.6% is not clear.

This sentence has completely been rewritten in the revised manuscript.

Also, the use of sulbactam did not improve the resistance is an important point and authors must use alternate ways to address it. Use of another B lactamase inhibitor in these experiments should suffice.

We can use only the 22 antimicrobial agents contain in the 96-well microtiter plates (CN3F, Sensititre, Thermo Scientific). For this reason, the use of alternate ways or alternate another B-lactamase inhibitor was not possible.

Overall, the purpose of this section, the objective and representation of their data in terms of an increase and decrease in the resistance is not clear and I suggest authors to rewrite this sentence after clearly thinking it through. The authors need to clearly mention the purpose and their method of measurement of resistance used in this study. What do the authors mean by a “break point” and why is it necessary?

The term “breakpoint” is defined in tables 1 and 2 and also in Material and Methods, section 4.2. Antimicrobial Sensisivity Testing. In addition, all the Results section has been rewritten and tables 1 and 2 have been modified, now highlighting susceptibilities instead of resistances. Besides, the title of the article has been improved for a better understanding.

The authors use of “worse” when less percentage of resistant isolates are obtained does not appear suitable for a scientific literature especially for this Journal which keeps publishing research aimed at reducing the incidents of antibiotic resistance.

The term “worse” has been omitted throughout the rewritten manuscript.

Tigecycline appears to be a very efficient drug as only 25% of the isolates show resistance to it. Any comments?

This sentence has been rewritten and therefore clarified in the rewritten manuscript.

Line 68 – Penicillin and Ampicillin are two different antibiotics. The authors must specify what they want to convey in the sentence.

This sentence has been modified in the rewritten manuscript and “Penicillin” has dissapeared of it.

Line 71 – “all animal isolates”? Please use complete sentences in the manuscript. It has been mentioned in the beginning of the section that animal isolates are being discussed. The repetition here is confusing. The authors must clearly present results from microorganisms isolated from different sources.

This suggestion has also been taken into account (as all of them): human isolates have sufficiently been differentiated from animal isolates.

Line 120 – missing “of”

The sentence in which appered this mistake have properly been corrected.

I do not think that the random incidence of drug resistance can be represented as “rate of susceptibility.

This has also been corrected in the rewritten manuscript.

Line 175 – Ampicillin was strongly inactive is not an ideal combination

We agree and, therefore, this sentence has been completely rewritten.

Reviewer 2 Report

In Vitro Antimicrobial Susceptibilities of Francisella tularensis subsp holarctica Isolates from Tularemia Outbreaks That Occurred over the Last Two Decades in Spain

The authors have analyzed the antibiotic resistance properties of a large collection of Francisella tularensis from humans and animals over the last two decades and compared to data reported more than two decades ago.  Although there is not much novelty in the research, the comprehensive analyses provide important information that contribute to the research field. 

Here are some specific comments about the manuscript.

Page 2 line 66 and Page 3 line 108: “other control reference strains” and “two other human F. tularensis subsp. holarctica reference strains”.  Mention that these reference strains are described in Material and Methods.

Page 2 line 86 and other places: “breakpoint”.  For the less informed readers, please provide a brief definition of breakpoint. 

Table 1 and Table 2: “MIC (minimum inhibitory concentration).”  The full form of MIC should be mentioned the first time it appears in the text and not in the table titles.

Table 1and 2: The same information is repeated in column 1 and 2. 

Table 1: For amikacin why do column 1 and column 2 not match (64 Vs 32)?

Table 1 Column 6: Check all calculations.  For example, in row 4, what does 84.9% correspond to?  125/148 should be 84.5% while 126/148 should be 85.1%.  For amikacin, 3/148 = 2.0%

Table 1and 2, column 5: There is a * in the heading but there is no * in the footnotes to explain it.

Table 2: For Ciprofloxacin, how was the 7.6% calculated? 2/29 = 6.9% and 3/29 = 10.3%

Figure 1: The information presented in Figure 1 is also present in Table 3, so it is not clear what the purpose of Figure 1 is.  Also, since Figure 1 is positioned before Table 3, the obvious question that arises is actually answered in Table 1: Why are the spans of the time frames not constant?

Page 6 line 146: “ser up”.  I guess, the authors mean “set up”.

Table 3 Column 1: Write the column heading in the same way the data are presented. 

Change                    Antibiotic                       to                          Antibiotic:

                        (parameters tested)                                         parameters tested

Page 6 lines 150 -158: This whole paragraph should be rewritten because the meaning is not clear. 

Page 6 line 151: Change “all except for cefoxitin” to “for all except cefoxitin”

Page 6 line 152: “to which were resistant”. Grammatically incorrect.  Maybe change to “to which they were resistant”

Page 6 line 152: “covered a group of”. Not clear what this means.

Page 6 line 154: “were able to grow from 13 to 17 antimicrobials”.  Not clear what this means.  Maybe the authors mean, “were able to grow in the presence of 13 to 17 antimicrobials”. 

Page 6 line 154: “Table 3” Do you mean Table 4?

Page 7 Table 4: “animal filed isolates”.  Not clear what “filed” means here.  Maybe the aiuthors mean, “isolates from animals”

Page 7 line 177: Change “no susceptible isolates to ampicillin were” to “no isolate susceptible to ampicillin was”

Page 7 line 177-180: “The three…in some cases”.  This sentence makes no grammatical sense.  Please rewrite.  Not clear, which “three other penicillins” are being discussed.  Not clear how penicillins can increase ampicillin activity or what it means.

Page 7 line 181: What is meant by “better”? Was the MIC lower or higher?

Page 7 line 183: “did not increase throughout this time”.  Not clear how “better” MIC means this.

Page 7 line 186: “seven compounds compared”. Mention where these results are in the manuscript.

Page 7 line 186: Change “any….not effective” to “none….effective”

Page 8 line 193: Change “of the Japanese” to “of Japanese”

Page 8 line 196: “were again strongly ineffective” Mention which table shows this result.

Page 8 line 225: Change “of authors” to “of other authors”

Page 8 line 228: “susceptibility to”.  I guess the authors mean “susceptibility for”

Page 8 line 237: Change “basis of the promising resistance” to “basis of the resistance”

Page 8 line 242: Change “can be” to “it can be”

Page 8 line 245: Change “No explanation has been able to find” to “We are unable to find an explanation”

Page 9 line 263: Change “thse” to “these”

Page 9 line 283: Please make correction “dehydrated 96-well microtiter plates”.  The plates were not dehydrated.

Page 9 line 292: Change “minimal inhibition concentration” to “minimum inhibitory concentration”

Page 10 line 310: Change “tetracicycline” to “tetracycline”

Page 10 line 313: Change “this five antibiotic” to “these five antibiotics”

Author Response

ANSWERS TO REVIEWER 2

In Vitro Antimicrobial Susceptibilities of Francisella tularensis subsp holarctica Isolates from Tularemia Outbreaks That Occurred over the Last Two Decades in Spain

The authors have analyzed the antibiotic resistance properties of a large collection of Francisella tularensis from humans and animals over the last two decades and compared to data reported more than two decades ago.  Although there is not much novelty in the research, the comprehensive analyses provide important information that contribute to the research field. 

Here are some specific comments about the manuscript.

Page 2 line 66 and Page 3 line 108: “other control reference strains” and “two other human F. tularensis subsp. holarctica reference strains”.  Mention that these reference strains are described in Material and Methods.

We have already indicated that these reference strains are described in M&M.

Page 2 line 86 and other places: “breakpoint”.  For the less informed readers, please provide a brief definition of breakpoint. 

The term “breakpoint” is defined in tables 1 and 2 and also in Material and Methods, section 4.2. Antimicrobial Sensisivity Testing.

Table 1 and Table 2: “MIC (minimum inhibitory concentration).”  The full form of MIC should be mentioned the first time it appears in the text and not in the table titles.

We have taken into account this suggestion in the rewritten manuscript.

Table 1 and 2: The same information is repeated in column 1 and 2. 

The information is not repeated: In column 1 appears the concentration range of each antimicrobial agent while in columna 2 appears the range that encompasses from the lowest to the highest concentration for each antimicrobial agents. Both can coincide but not necessarily.

Table 1: For amikacin why do column 1 and column 2 not match (64 Vs 32)?

Both columns do not match for the reason explained in the previous question.

Table 1 Column 6: Check all calculations.  For example, in row 4, what does 84.9% correspond to?  125/148 should be 84.5% while 126/148 should be 85.1%.  For amikacin, 3/148 = 2.0%

Ok, the calculations have all been checked and in the rewritten manuscript the percentages are referred to susceptibility of isolates, not to resistance of isolates.

Table 1and 2, column 5: There is a * in the heading but there is no * in the footnotes to explain it.

This mistake has already been corrected and “*” has been used to define what “breakpoint” is.

Table 2: For Ciprofloxacin, how was the 7.6% calculated? 2/29 = 6.9% and 3/29 = 10.3%

Thanks. This mistake has been corrected in the revised manuscript.

Figure 1: The information presented in Figure 1 is also present in Table 3, so it is not clear what the purpose of Figure 1 is.  Also, since Figure 1 is positioned before Table 3, the obvious question that arises is actually answered in Table 1: Why are the spans of the time frames not constant?

Figure 1 has been deleted in the rewritten manuscript. In addition, the span of the time frames are not constant because we try to approximate the number of F. tularensis isolates, not the the spans of the time frames in order that the statitical análisis were more reliable. This fact is explained in M&M, section 4.3.

Page 6 line 146: “ser up”.  I guess, the authors mean “set up”.

Thanks, effectively. We want to writte “set up”, not “ser up”.

Table 3 Column 1: Write the column heading in the same way the data are presented. 

Change                    Antibiotic                       to                          Antibiotic:

                        (parameters tested)                                         parameters tested

 We have taken into account your suggestion and “()” have been removed in the revised version of our manuscript.

Page 6 lines 150 -158: This whole paragraph should be rewritten because the meaning is not clear. 

Certainly, this paragraph was very confused and it has entirely rewritted.

Page 6 line 151: Change “all except for cefoxitin” to “for all except cefoxitin”

This has already been corrected in the rewritten manuscript.

Page 6 line 152: “to which were resistant”. Grammatically incorrect.  Maybe change to “to which they were resistant”

 This has already been corrected in the rewritten manuscript.

Page 6 line 152: “covered a group of”. Not clear what this means.

This expression has already been sorted up in the revised manuscript.

Page 6 line 154: “were able to grow from 13 to 17 antimicrobials”.  Not clear what this means.  Maybe the authors mean, “were able to grow in the presence of 13 to 17 antimicrobials”. 

Yes, of course. You have reason. This expression has already been corrected in the rewritten manuscript.

Page 6 line 154: “Table 3” Do you mean Table 4?

Yes, we have renumered Table 3 as Table 4 in the revised manuscript.

Page 7 Table 4: “animal filed isolates”.  Not clear what “filed” means here.  Maybe the aiuthors mean, “isolates from animals”

Yes, the authors meant “isolates from animals”. This is already modified in the revised manuscript.

Page 7 line 177: Change “no susceptible isolates to ampicillin were” to “no isolate susceptible to ampicillin was”

This change has already been carried out in the revised manuscript.

Page 7 line 177-180: “The three…in some cases”.  This sentence makes no grammatical sense.  Please rewrite.  Not clear, which “three other penicillins” are being discussed.  Not clear how penicillins can increase ampicillin activity or what it means.

We agree. This sentence was very confused and it has been rewritten to clarify it in the second version of our manuscript.

Page 7 line 181: What is meant by “better”? Was the MIC lower or higher?

This word was also confused in this context. We have rewritten that sentence to clarify.

Page 7 line 183: “did not increase throughout this time”.  Not clear how “better” MIC means this.

The same: This word was also confused in this context. We have rewritten that sentence to clarify.

Page 7 line 186: “seven compounds compared”. Mention where these results are in the manuscript.

This suggestion has already been taken into account in our revised manuscript.

Page 7 line 186: Change “any….not effective” to “none….effective”

This change has already been carried out in the second version of our manuscript.

Page 8 line 193: Change “of the Japanese” to “of Japanese”

“The” has been deleted in the new version of our manuscript.

Page 8 line 196: “were again strongly ineffective” Mention which table shows this result.

This sentence has been rewritten because it was confused in the original manscript.

Page 8 line 225: Change “of authors” to “of other authors”

This change has already been carried out in the revised manuscript.

Page 8 line 228: “susceptibility to”.  I guess the authors mean “susceptibility for”

Yes, certainly. We meant “susceptibility for”. This change has also been carried out.

Page 8 line 237: Change “basis of the promising resistance” to “basis of the resistance”

“Promising” has been deleted in the rewritten manuscript.

Page 8 line 242: Change “can be” to “it can be”

“it” has been included as subject in that sentence.

Page 8 line 245: Change “No explanation has been able to find” to “We are unable to find an explanation”

This change has been carried out in the rewritten manuscript.

Page 9 line 263: Change “thse” to “these”

This typographic mistake has already been corrected.

Page 9 line 283: Please make correction “dehydrated 96-well microtiter plates”.  The plates were not dehydrated.

We agree. This sentence has properly been rewritten in the second version of our manuscript (M&M, section 4.2).

Page 9 line 292: Change “minimal inhibition concentration” to “minimum inhibitory concentration”

This mistake has already been corrected.

Page 10 line 310: Change “tetracicycline” to “tetracycline”

This typographic mistake has already been corrected.

Page 10 line 313: Change “this five antibiotic” to “these five antibiotics”

This typographic mistake has already been corrected.
